# High-Level Production of Soluble Cross-Reacting Material 197 in *Escherichia coli* Cytoplasm Due to Fine Tuning of the Target Gene’s mRNA Structure

**DOI:** 10.3390/biotech12010009

**Published:** 2023-01-11

**Authors:** Yulia Alexandrovna Khodak, Alexandra Yurievna Ryazanova, Ivan Ivanovich Vorobiev, Alexander Leonidovich Kovalchuk, Nikolay Nikolaevich Ovechko, Petr Gennadievich Aparin

**Affiliations:** 1Federal State Institution, Federal Research Centre “Fundamentals of Biotechnology”, The Russian Academy of Sciences, 119071 Moscow, Russia; 2National Research Center, Institute of Immunology Federal Medical-Biological Agency of Russia, 115522 Moscow, Russia; 3ATVD-TEAM Co., Ltd., 115522 Moscow, Russia

**Keywords:** CRM197, carrier protein, protein solubility, protein expression, *Escherichia coli*, SHuffle strain

## Abstract

Cross-reacting material 197 (CRM197) is a non-toxic mutant of the diphtheria toxin and is widely used as a carrier protein in conjugate vaccines. This protein was first obtained from the supernatant of the mutant *Corynebacterium diphtheriae* strain. This pathogenic bacteria strain is characterized by a slow growth rate and a relatively low target protein yield, resulting in high production costs for CRM197. Many attempts have been made to establish high-yield protocols for the heterologous expression of recombinant CRM197 in different host organisms. In the present work, a novel CRM197-producing *Escherichia coli* strain was constructed. The target protein was expressed in the cytoplasm of SHuffle T7 *E. coli* cells without any additional tags and with a single potential mutation—an additional Met [−1]. The fine tuning of the mRNA structure (the disruption of the single hairpin in the start codon area) was sufficient to increase the CRM197 expression level several times, resulting in 150–270 mg/L (1.1–2.0 mg/g wet biomass) yields of pure CRM197 protein. Besides the high yield, the advantages of the obtained expression system include the absence of the necessity of CRM197 refolding or tag removal. Thus, an extensive analysis of the mRNA structure and the removal of the unwanted hairpins in the 5′ area may significantly improve the target protein expression rate.

## 1. Introduction

Cross-reacting material 197 (CRM197) is a non-toxic mutant of diphtheria toxin (DT), which was obtained in 1972 through nitrosoguanidine-induced mutagenesis [1]. The lack of toxicity in CRM197 is due to Gly substitution with Glu in position 52 (G52E) inside the enzyme’s active center [2]. CRM197 is widely used as a carrier protein in conjugate vaccines (Table 1) [3,4,5,6]. It can also be utilized as a boosting antigen for immunization against diphtheria [7,8,9]. The worldwide CRM197 manufacturing capability falls behind the market demands. Therefore, many research groups have been developing new protocols for high-yield CRM197 expression in different host organisms during last two decades.

The expression system for DT and CRM197 implies the cultivation of corresponding *Corynebacterium diphtheriae* strains, which is the natural host of DT and the causative agent of diphtheria. CRM197 or DT is secreted into culture medium and then purified from the clarified broth through several chromatographic steps. The best reported yield of the target protein using this expression system is in the range of 175–250 mg/L [10].

CRM197 expression in *C. diphtheriae* has some obvious disadvantages: the necessity of a biosafety level 2 (BSL-2) containment facility and an inevitable small risk of a reverse mutation converting CRM197 into DT and restoring the toxicity of *C. diphtheriae*. At the same time, *C. diphtheriae* grows slower than *Escherichia coli*. Taken together, these facts lead to the high cost of purified CRM197 and encourage the search for alternative expression hosts.

The heterologous expression of recombinant CRM197 was tested in a broad range of organisms such as *E. coli* [11,12,13,14,15,16,17,18,19,20], *Bacillus subtilis* [21,22], *Pseudomonas fluorescens* [23], attenuated *Salmonella typhi* [24], and *Pichia pastoris* [25], along with low-alkaloid tobacco plants and carrot cell cultures [26]. Three expression systems were successful enough to serve as a basis for large-scale manufacturing. The resulting CRM197 proteins are distributed as PeliCRM™ (Ligand Pharmaceuticals, previously sold by Pfenex, expression in *P. fluorescens*) [23], EcoCRM™ (Fina BioSolutions LLC, expression in *E. coli*) [19], and simply CRM197 (Scarab Genomics LLC, expression in Clean Genome^®^ *E. coli*) [20]. None of these expression protocols has been reported in peer-reviewed articles. The recombinant variants of CRM197 expressed in *E. coli* and *P. fluorescens* were shown to be highly similar to those expressed in *C. diphtheriae* and sometimes even more pure [27].

*E. coli* is undoubtedly the most popular host microorganism for the expression of heterologous proteins, both in academic research and in biotechnology. Its well-known advantages are the following: (1) *E. coli* is able to grow fast and form high-cell-density cultures on relatively inexpensive growth media; (2) the *E. coli* genome is completely sequenced, and its genetics and metabolism are well studied; (3) a variety of *E. coli* strains and suitable expression vectors have been developed; and (4) the construction of new *E. coli* strains is also not complicated. *E. coli* has been extensively used for the large-scale manufacturing of therapeutic proteins that do not require glycosylation [28]. Although CRM197 expression in *E. coli* solves the problem of the cell culture growth rate and eliminates the necessity of BSL-2 containment requirements, increasing the protein yield remains an issue. Moreover, new complications arise, namely the solubility of the obtained CRM197 and its purity after purification from cell lysate (in contrast to purification from the culture broth of *C. diphtheriae*).

The present work reports a high-yield CRM197 expression system based on *E. coli*. The biosynthesis of CRM197, primarily in the soluble monomeric form, is provided by two main factors: (1) the detailed engineering of the mRNA structure around the start codon of the target gene and (2) the usage of the SHuffle^®^ T7 strain (New England Biolabs, Ipswich, MA, USA), which has cytoplasm with a diminished reducing potential and contains the re-targeted and overexpressed DsbC disulfide isomerase. The constructed expression host proved to grow well under high-cell-density conditions. The soluble CRM197 was purified through several chromatographic steps. The resulting yield was among the best values reported in peer-reviewed articles (Table 2).

## 2. Materials and Methods

### 2.1. Generation of the CRM197-Encoding Gene

The nucleotide sequence encoding DT (GenBank: X00703.1) was modified as follows: A nucleotide replacement was made, which provided Gly52Glu substitution, turning DT into the non-toxic CRM197. The CRM197 gene was shortened, resulting in the removal of the natural signal sequence in the encoded protein (25 amino acid residues on the DT N-terminus). The ATG codon was added upstream of the mature CRM197, targeting the protein to the cytoplasm. The resulting nucleotide sequence was optimized based on *E. coli* codon frequency and the potential RNA secondary structure using the Codon Optimization Tool (https://eu.idtdna.com/pages/tools/codon-optimization-tool, accessed on 23 December 2022).

### 2.2. Molecular Cloning

The CRM197-encoding gene was synthesized via the ligation of preformed DNA duplexes consisting of 5′-phosphorylated overlapping oligonucleotides (Evrogen, Moscow, Russia). Then, it was amplified by polymerase chain reaction (PCR) and ligated into a precut pAL2-T vector (Evrogen, Moscow, Russia). The resulting nucleotide sequence was sequence-verified and subcloned into the expression plasmids pET28a (Novagen, Merck, Germany) and pHYP [29] using the NcoI and XhoI restriction sites. pHYP is a higher-copy-number derivative of the pET28a plasmid, which bears the toxin–antitoxin *hok*/*sok* (*ParB*) genetic element that prevents plasmid elimination [30,31]. The resulting plasmids, pET28a-CRM197-gcc and pHYP-CRM197-gcc (Table 3), were sequence-verified again and transformed into the *E. coli* BL21(DE3) [32] and SHuffle^®^ T7 [33] (New England Biolabs, Ipswich, MA, USA) strains using the heat shock method [34].

### 2.3. Mutagenesis

The CRM197-encoding gene variant with a leader sequence for periplasmic expression was obtained by PCR using the pHYP-CRM197-gcc plasmid as a template and primers AD-GV-FLGC-F and AD-GV-C-R (Appendix A). The PCR products were cloned into the pAL2-T vector, sequence-verified, and subcloned into the expression vector pHYP, resulting in the pHYP-CRM197-peri plasmid (Table 3).

The deletion derivatives of the CRM197-encoding gene were obtained through PCR using the pHYP-CRM197-gcc plasmid as a template and a set of primers listed in Appendix A. The PCR products were cloned into the pAL2-T vector, sequence-verified, and subcloned into the expression vector pHYP, resulting in the pHYP-D1, pHYP-D2, and pHYP-D3 plasmids (Table 3).

The single-nucleotide substitutions in the third codon of the CRM197-encoding gene were obtained, as described above, using a set of primers listed in Appendix A. The resulting plasmids were named pHYP-CRM197-gca, pHYP-CRM197-gcg, pHYP-CRM197-gct, pHYP-CRM197-acc, pHYP-CRM197-tcc, and pHYP-CRM197-ccc (Table 3).

### 2.4. RNA Secondary Structure Analysis

The secondary structures of the CRM197-encoding mRNA variants were predicted using the RNAfold [35] and SPOT-RNA [36] algorithms. First, the full-length mRNA sequences were analyzed with both online tools. Then, the input mRNA sequences were shortened and analyzed again with RNAfold to find a minimal part that would form a relatively independent secondary structure element.

### 2.5. Analytical Gene Expression

The *E. coli* BL21(DE3) or SHuffle^®^ T7 cells bearing a CRM197-encoding plasmid were inoculated into 5 mL of 2× YT medium with 1% glucose and 50 mg/L kanamycin in test tubes. After an overnight cultivation at 37 °C at 180 rpm, the culture was diluted 50-fold with fresh 2× YT medium containing 0.1% glucose and 50 mg/L kanamycin. The cultivation continued in test tubes at 37 °C until an optical density at 600 nm (OD_600_) of 0.6–0.8 was reached. Then, the flasks were cooled to 22–37 °C, and isopropyl β-D-1-thiogalactopyranoside (IPTG) was added to a final concentration of 1 mM. The cultivation continued overnight at 22–37 °C.

For CRM197 quantification, aliquots were taken from the cell cultures and centrifuged. The supernatants were discarded; the precipitates were resuspended in lysis buffer (20 mM Tris–HCl, 2 mM ethylenediaminetetraacetic acid (EDTA), 1 mM phenylmethylsulfonyl fluoride (PMSF), pH 7.5) and lysed by sonication on ice. Aliquots of the lysates were analyzed by Laemmli electrophoresis in order to determine total (soluble + insoluble) amount of CRM197. Then, the lysates were centrifuged at 12,000× *g* for 20 min. The clarified supernatants were analyzed by Laemmli electrophoresis to determine the amount of soluble CRM197.

### 2.6. Kinetics of Gene Expression at Different Temperatures

The *E. coli* SHuffle^®^ T7 cells bearing a CRM197-encoding plasmid were inoculated into 5 mL of 2× YT medium with 1% glucose and 50 mg/L kanamycin and grown overnight at 37 °C. Three flasks with 100 mL of 2× YT medium with 1% glucose and 50 mg/L kanamycin were inoculated with 2 mL of the overnight seed culture and grown at 37 °C until OD_600_ reached 0.8. The flasks were then cooled to 22 °C inside the shaker for approximately 20 min and induced by adding IPTG to a final concentration of 1 mM. The cultivation continued at 22, 27, 32, or 37 °C. Aliquots (1 mL) were taken 1, 2, 3, 5, 7, and 20 h after the induction. The OD_600_ value was measured for each sample. The cell suspensions were then centrifuged in preweighed test tubes, the supernatant was completely removed, and the wet cell pellet was weighed again and stored frozen for further analysis.

### 2.7. Cell Cultivation in a Fermenter

High-cell-density cultivation was performed in fed-batch mode. A 16-L fermenter containing 10 L of the complex semi-defined medium described in [14] with 50 mg/L kanamycin sulfate was inoculated with 100 mL of overnight culture. Cultivation was performed at 28 °C with 0.5 bar overpressure and 800 L/h sparged air consumption. The dissolved oxygen (DO) was kept at 20% (regulated by the stirring speed); the pH was maintained at 7.0 ± 0.1 using a 5% aqueous ammonia solution. Laprol anti-foam (Scientific production association “Macromer”, Vladimir, Russia) was added when necessary. After glucose exhaustion (seen as a sharp increase in the DO level), the operation mode was switched to fed-batch using a concentrated glycerol-based feed solution [14]. When the OD_600_ exceeded 40, the temperature was decreased to 23 °C and IPTG was added to a final concentration of 1 mM. The cultivation was completed when the OD_600_ stabilized. The cell paste was obtained by centrifugation at 13,000× *g* for 10 min and stored at −70 °C.

### 2.8. Refolding of the Inclusion Body Fraction

The wet biomass was resuspended at 1:9 (*w*/*v*) in buffer A (50 mM Tris–HCl, pH 8.0, 0.1% Triton X-100, 2 mM EDTA, and 20 mkg/mL lysozyme) and incubated for 30 min at 4 °C. The suspension was sonicated until its viscosity dropped and then centrifuged. The precipitate was resuspended at 1:1 (*w*/*v*) in buffer B (50 mM Tris–HCl, pH 8.0, 3 M urea, 500 mM NaCl, and 2 mM EDTA), sonicated until the viscosity drop, incubated for 10 min at 4 °C with constant stirring, and centrifuged again. The precipitate was resuspended at 1:4 (*w*/*v*) in buffer C (10 mM Tris–HCl, pH 8.0, and 1 mM EDTA) and centrifuged. The resulting precipitate represented purified inclusion bodies; it could be used for refolding experiments immediately or stored at –20 °C.

The purified inclusion bodies were resuspended in buffer D (20 mM Tris–HCl, pH 8.0, 8 M urea, and 10 mM DTT) up to 10% concentration and kept for 1 h at 37 °C with constant stirring. Then, the suspension was centrifuged at 12,000× *g* for 20 min, and the supernatant was diluted 40-fold with the refolding solution (20 mM Tris–HCl, pH 7.5, 30 mM NaCl, 0.1 mM glutathione disulfide, and 1 mM reduced glutathione) and incubated at 4 °C overnight with constant stirring. The resulting solution of refolded CRM197 was clarified by centrifugation and used for analysis or purification, as described below.

### 2.9. Isolation of Periplasmic Proteins

Wet biomass (1 g) was resuspended in 80 mL of hypertonic buffer (20% sucrose, 5 mM EDTA, 30 mM Tris–HCl, pH 8.0), incubated for 10 min at room temperature, and centrifuged at 20,000× *g* for 10 min. The supernatant (hypertonic fraction) was discarded. The precipitate was resuspended in a cold (+4 °C) hypotonic solution (5 mM MgSO_4_), incubated on ice for 20 min to release periplasmic CRM197, and centrifuged at 20,000× *g* for 10 min. The amount of periplasmic CRM197 in the hypotonic fraction was analyzed by Laemmli electrophoresis. The precipitate was resuspended in 20 mM Tris–HCl (pH 8.5), sonicated on ice, and analyzed by Laemmli electrophoresis in order to quantify the residual amount of cytoplasmic CRM197.

### 2.10. CRM197 Purification

The harvested cells were resuspended in lysis buffer (20 mM Tris–HCl, 2 mM EDTA, 1 mM PMSF, pH 7.5) and then disrupted by sonication without the addition of any detergents. The lysate was cleared by centrifugation. The ionic strength of the supernatant was reduced by diafiltration. The diafiltered solution was applied to a Capto Q anion-exchange column (Cytiva, Marlborough, MA, USA); the target protein was eluted with an increased NaCl concentration (150 mM). The resulting eluate was loaded onto a ceramic hydroxyapatite column (Bio-Rad Laboratories, Hercules, CA, USA) and eluted with a high-phosphate buffer. The obtained solution was applied to a Capto MMC column (Cytiva, Marlborough, MA, USA) and eluted with a solution that had both high pH and increased ionic strength. The resulting eluate was concentrated and buffer-exchanged into 100 mM potassium phosphate (pH 7.2) using ultrafiltration and diafiltration. The concentrated CRM197 was stored at −70 °C.

### 2.11. Laemmli Electrophoresis

Polyacrylamide gel electrophoresis (PAGE) or Laemmli electrophoresis [37] were performed in 60 × 80 × 1 mm^3^ gels using Mini-PROTEAN Tetra Cell (Bio-Rad Laboratories, Hercules, CA, USA) in Tris–glycine buffer (25 mM Tris, 192 mM glycine, and 0.1% sodium dodecyl sulfate (SDS)). The separating gel contained 9.7% acrylamide and 0.3% *N*,*N*′-methylenebis(acrylamide). The concentrating gel contained 3.9% acrylamide and 0.1% *N*,*N*′-methylenebis(acrylamide). Before applying to the gel, each specimen was mixed with a 4× loading buffer (250 mM Tris–HCl, pH 8.5, 8% SDS, 1.6 mM EDTA, 0.024% pyronin Y, 0.04% bromophenol blue, and 40% glycerol) and incubated for 5 min at 95 °C. The samples were analyzed under reductive or non-reductive conditions, i.e., the loading buffer contained or lacked 400 mM dithiothreitol (DTT). The PageRuler Unstained Protein Ladder and the PageRuler Prestained Protein Ladder (#26614 and #26616, ThermoFisher Scientific, Waltham, MA, USA) were used as molecular weight markers. Gels were stained with colloidal Coomassie Brilliant Blue dye [38]. The stained gels were scanned with a flatbed scanner in the transparent mode as 16-bit grayscale images. Densitometry was performed with TotalLab TL120 software (Nonlinear Dynamics Ltd., Newcastle, UK) using a linear calibration curve made from known amounts of purified CRM197. The statistical analysis was conducted using a one-way ANOVA followed by Bonferroni’s post hoc comparisons tests (https://astatsa.com/OneWay_Anova_with_TukeyHSD, accessed on 23 December 2022).

### 2.12. Western Blotting

The transfer of proteins from a polyacrylamide gel to a nitrocellulose membrane (GVS S.p.A., Bologna, Italy) was performed overnight in a Tris–glycine buffer using a TE 70 PWR semi-dry transfer unit (Cytiva, Marlborough, MA, USA). The membrane was blocked with a 1% bovine serum albumin (BSA) solution in phosphate-buffered saline (PBS: 137 mM NaCl, 2.7 mM KCl, 10 mM Na-phosphate, pH 7.3–7.5). Two different murine monoclonal IgG1 antibodies against DT (MBS310410 and MBS5305972, MyBioSource, San Diego, CA, USA) were diluted to 1:2000 with 1% BSA in PBS and used as primary antibodies. Goat polyclonal IgG against murine IgG (ab6789, Abcam, Cambridge, MA, USA) conjugated with horseradish peroxidase (HRP) was diluted to 1:2000 with 1% BSA in PBS and used as a secondary antibody. The membrane was stained with 0.04% 3,3′-diaminobenzidine (DAB) and a 0.1% H_2_O_2_ solution in PBS enhanced with 0.04% CoCl_2_ and 0.04% NiCl_2_ [39].

### 2.13. Native Gel Electrophoresis

Native gel electrophoresis was performed in 60 × 80 × 1 mm^3^ gels using a Mini-PROTEAN Tetra Cell (Bio-Rad Laboratories, Hercules, CA, USA) in TBE buffer (90 mM Tris, 90 mM boric acid, and 2 mM EDTA). The gel contained 8.7% acrylamide and 0.3% *N*,*N*′-methylenebis(acrylamide). Each specimen was mixed with a 4× loading buffer (250 mM Tris–HCl, pH 8.5, 1.6 mM EDTA, 0.04% bromophenol blue, and 40% glycerol) without preheating before applying to the gel. The gels were stained with colloidal Coomassie Brilliant Blue dye [38].

### 2.14. Analytical Size-Exclusion Chromatography

A Waters chromatography system with a 2795 separation module and a 2996 photodiode array detector (Waters Corporation, Milford, MA, USA) was used for analytical size-exclusion chromatography. The analysis was performed using a Superdex 200 10/300 GL column (Cytiva, Marlborough, MA, USA) with a flowrate of 0.5 mL/min in PBS buffer containing 0.02% NaN_3_. Each sample was injected in a volume of 30 µL. A Gel Filtration Calibration Kit HMW (Cytiva, Marlborough, MA, USA) was used for column calibration.

### 2.15. N-Terminal Protein Sequencing

A PPSQ-33A protein sequencer (Shimadzu Corp., Kyoto, Japan) was used to determine the N-terminal protein sequence by Edman degradation combined with high-performance liquid chromatography (HPLC). The analysis was performed according to the standard manufacturer’s protocol.

## 3. Results

### 3.1. Gene and Vector Construction

The synthetic gene encoding CRM197 was created using the codon optimization algorithm; an *E. coli* codon frequency table was utilized. The natural N-terminal signal sequence was removed, and the ATG start codon was added upstream to the sequence encoding the mature protein. The potential mRNA secondary structures were also optimized (see Materials and Methods). The synthetic gene was cloned into two different pET-derived vectors, namely pET28a (Novagen, Merck, Germany) and pHYP [29]. The resulting plasmids were called pET28a-CRM197-gcc and pHYP-CRM197-gcc, respectively. Both vectors carried a kanamycin resistance gene, a T7 promoter, and a lac operator, which allowed inducing the expression of the target protein by the addition of IPTG. The pHYP plasmid also contained the toxin–antitoxin genetic element *hok*/*sok* (*ParB*), which was expected to kill the bacteria immediately if they eliminated the plasmid, potentially increasing the productive induction time. 

### 3.2. Refolding of CRM197 from Inclusion Bodies

When expressed in the *E. coli* BL21(DE3) strain, CRM197 was almost completely insoluble and formed inclusion bodies. The inclusion bodies were purified and then solubilized using 8 M urea. In order to stimulate the protein refolding, the samples were diluted 40-fold with a glutathione-based solution and kept at 4 °C overnight. Similar procedures were described in articles published to date [11,16,17], but they were not accompanied by ion-exchange chromatographic purification. Up to 70% of CRM197 remained soluble upon refolding, according to the SDS-PAGE analysis, but the refolded protein did not bind to the anion-exchange resin under the low-salt conditions that are standard for the purification of CRM197 and DT [10]. Analytical size-exclusion chromatography showed the absence of monomeric CRM197 in solution after the refolding procedure. Moreover, we found that the refolded CRM197 was mostly denatured during the ammonium sulfate precipitation at 50% saturation and remained insoluble when the precipitate was resuspended in a low-salt solution. Thus, another expression strategy had to be found in order to obtain the target protein in a soluble monomeric form.

### 3.3. Periplasmic Expression of CRM197

The expression of CRM197 in the periplasmic space of the *E. coli* B derivative strains described in [14] was reproduced by adding the sequence encoding the FlgI leader peptide upstream of the CRM197 open reading frame. The resulting plasmid was used for the transformation of the BL21(DE3) cells and the expression of CRM197 in the periplasmic space in shake flasks. The periplasmic fraction was obtained by cell treatment with hypertonic and hypotonic solutions and used for CRM197 purification by ion-exchange chromatography. The target protein was present in a soluble form in the periplasm of the BL21(DE3) cells, according to the SDS-PAGE analysis, which was in good agreement with [14]. At the same time, the purification of CRM197 from the periplasmic fraction or the total cell lysate by ion-exchange chromatography also resulted in a low yield of the purified product: up to 0.13 mg/g of wet biomass, far below the expected multi-gram per liter level described in [14].

Thus, the only viable option for high-level soluble and monomeric CRM197 expression in *E. coli* cells is expected to be the accumulation of properly oxidized products in the cytoplasm.

### 3.4. Choice of Expression Strain and Vector

The proper three-dimensional structure of CRM197 and DT requires the formation of two intramolecular disulfide bonds. On the contrary, the *E. coli* cytoplasm has a reductive redox potential that promotes the disruption of disulfide bonds. This is a common problem when expressing heterologous proteins in *E. coli* [28]. To cope with this problem, special *E. coli* strains have been developed [40]. One of them is SHuffle™ (New England Biolabs, Ipswich, MA, USA), which carries deletions of trxB and gor (genes coding for thioredoxin reductase and glutathione reductase, respectively). It also carries mutated peroxidase AhpC* (gaining the activity of disulfide bond reductase) and overexpresses a cytosolic version of the disulfide bond isomerase DsbC [33]. Thus, SHuffle™ cells have cytoplasm with diminished reducing potential and can also correct mis-oxidized pairs of cysteine residues in proteins.

In order to express recombinant CRM197 in a soluble form, the plasmids pET28a-CRM197-gcc and pHYP-CRM197-gcc were transformed into the *E. coli* SHuffle^®^ T7 strain. Although some amount of CRM197 was found in inclusion bodies, as before, most of the target protein was present in the soluble protein fraction (3 h after the IPTG addition). The CRM197 expression level was determined by gel densitometry, with pure CRM197 as the standard; it was 12% higher for the pHYP-CRM197-gcc plasmid than for the pET28a-CRM197-gcc plasmid (Figure 1A). Therefore, pHYP-CRM197-gcc was used as the expression vector in further experiments.

### 3.5. Optimization of Expression Temperature

Besides the diminished cytoplasmic reducing potential, some other conditions during the expression of a protein can influence its solubility. One of the relevant variables is the cultivation temperature. Several works reported enhancing the fraction of soluble CRM197 upon decreasing the expression temperature [12,14]. This observation can be explained by slowing the total rate of protein biosynthesis at lower temperatures, thus providing more time for correct CRM197 folding.

We found that expression at 37 °C yielded no soluble CRM197, according to the SDS-PAGE analysis. Then, we tested the CRM197 expression levels in shake flasks at three different decreased temperatures: 32, 27, and 22 °C. While the biomass accumulation over time was similar in all cases (Figure 1B), the concentration of soluble CRM197 increased over several hours at different rates and then dropped after the overnight incubation (Figure 1C). The highest content of soluble CRM197 was detected 7 h after induction at 27 °C. However, the expression level did not exceed 30 mg of soluble CRM197 per 1 L of culture medium. This value is an order of magnitude lower than the typical results of CRM197 expression in *C. diphtheriae* culture. Thus, further optimization of the expression system was necessary.

### 3.6. Mutagenesis of mRNA

Although the CRM197-encoding gene was initially optimized taking into account the potential mRNA secondary structure, the presence of some hairpins interfering with ribosome binding, assembly, and elongation was still possible. It is widely known that the gene translation rate can be influenced by the mRNA structure, especially in the 5′ untranslated region and around the start codon. This fact can be demonstrated by significant changes in gene expression levels upon mutations or deletions in the aforementioned region, which disrupt the local elements of the mRNA secondary structure. For example, the deletion of a short nucleotide fragment encoding several N-terminal amino acid residues dramatically increased the expression level of papillomavirus capsid proteins in *E. coli* [41,42]. A similar effect could be expected in the case of CRM197 expression.

A set of plasmids was constructed (Table 3), where the gene encoding CRM197 lacked the first one (pHYP-D1), two (pHYP-D2), or three (pHYP-D3) codons corresponding to the N-terminal amino acid residues (downstream of the ATG start codon). These short deletions resulted in significant increases of the total CRM197 expression level: four-fold and three-fold when one and two amino acid residues were deleted, respectively. However, the expression level returned to the initial value upon the deletion of three N-terminal amino acid residues (Figure 2A). A similar tendency was observed for the soluble CRM197; the concentration increased when one or two N-terminal residues were deleted and returned to the initial value in the case of a triple deletion.

The three-fold increase in the amount of soluble CRM197 upon the deletion of the first two N-terminal amino acid residues (six nucleotides) pointed to a particular importance of the second coding triplet for the expression level of this gene. For a more detailed investigation, we constructed another set of plasmids without any deletions but with all possible substitutions in the first and third positions of the second triplet (Table 3). The nucleotide changes in the first position led to Ala substitutions with Thr, Ser, or Pro, while the changes in the third position were synonymous and retained Ala in the target protein.

All three variants of synonymous substitution increased the total CRM197 expression level four-fold (Figure 2B). Thus, the assumption about the influence of the mRNA structure on the translation rate seems highly likely. The Ala substitutions with Ser and Thr increased the total CRM197 expression level four- and five-fold, respectively (Figure 2B), while the substitution with Pro resulted in a six-fold decrease. A quantitative analysis of soluble CRM197 (Figure 2C) revealed similar effects; the proportions of the soluble and insoluble CRM197 were similar for all tested codon variants.

Besides the mRNA structure, the gene expression level can be limited by the low frequency of the required tRNAs. Out of the seven tested codons, only one (CCC) is rare in *E. coli* (<0.5%); moreover, its tRNA also has very low abundance in *E. coli* [43]. This fact may explain the large drop in the CRM197 expression level when the CCC codon is used. However, the other six tested codons are not rare in *E. coli.* Furthermore, there are two tRNAs for Ala: one recognizes GCA, GCG, and GCU, while the other recognizes GCC and GCU [43]. Thus, the codon GCU in mRNA can bind to both tRNAs, which could lead to an elevated CRM197 expression level with this codon. Unfortunately, this was not the case (Figure 2B), and the leading hypothesis thus remains the one considering the mRNA secondary structure.

In order to find an explanation for the observed CRM197 biosynthesis variations, other than codon frequencies, we made a series of mRNA secondary structure calculations using the RNAfold [35] and SPOT-RNA [36] online tools. The RNAfold algorithm is well suited for the analysis of base pairing in relatively short RNA fragments (including the non-canonical G:U pair), while the SPOT-RNA program can predict the secondary structure of long RNA (up to 2000 bases), taking into account different variants of non-canonical base pairing and pseudoknots stabilized by spatial interactions.

Initially, the RNAfold algorithm was applied to the full-length target mRNA, i.e., from the supposed transcription start to the supposed transcription terminator (1825 bases: Figure 3A). A major part of each mRNA was predicted to participate in secondary structure elements; most of the hairpins were formed between fragments located close to each other (Appendix A). The region of interest was predicted to participate in long-range interactions, although with quite low probabilities (Appendix A). The optimal structures with the different codons indeed differed. However, the frequency of each optimal structure in the ensemble was assessed as 0.00% (Appendix A). Therefore, an attempt was made to test if the region of interest participated in any significant short-range interactions. The sequence for the analysis was trimmed down to 70 bases from the supposed transcription start to the start codon and a fragment of the same length downstream from the start codon (140 bases in total; Figure 3A). In the third iteration, only the hairpin containing the single-nucleotide changes was analyzed (33 bases). The AUG start codon was partially included in the hairpin. The hairpin structure was predicted in the case of the GCC codon with a higher probability (marked in red in Figure 3B) than in the other cases. However, the calculated values of minimum free energy for the optimal secondary structure as well as the calculated values of free energy for the thermodynamic ensemble (Appendix A) were poorly correlated with the experimentally obtained expression pattern. No visible differences were found for the “bad” GCC codon.

The SPOT-RNA algorithm is based on deep contextual learning and is trained using known high-resolution RNA spatial structures [36]. We used this program to analyze the mRNA structure from the supposed transcription start to the supposed transcription terminator (1825 bases). A difference was found between the sequence containing GCC codon and those containing GCT or GCG codons: canonical base-pairing was predicted between A_76_G_77_C_78_C_79_ and G_1298_G_1299_C_1300_U_1301_ fragments (numbering from the supposed transcription start) in the case of GCC codon, while in the case of GCT and GCG codons these nucleotides did not participate in any base-pairing (Figure 3C). The SPOT-RNA results for all the full-length mRNA variants are shown in Appendix A in the Appendix A. In the case of the GCA codon, canonical base-pairing was predicted between A_76_G_77_C_78_A_79_ and the extremely remote fragment U_1756_G_1757_C_1758_U_1759_. Since this interaction implies two A:U pairs and two G:C pairs, it is weaker than the case of the GCC codon (one A:U pair and three G:C pairs).

Thus, the differences in the experimentally obtained CRM197 expression patterns can be explained by a combination of different mechanisms: the lowest expression level in the case of the CCC codon resulted from the low tRNA abundance, while in the other cases the expression level could be limited by long-range interactions in the mRNA secondary structure. A target gene with any one of the synonymous codons GCA, GCG, and GCT could have been taken for further experiments since there was no significant difference in the expression level. We used the GCT codon, as the DT gene sequence deposited in GenBank (X00703.1) contained this variant.

### 3.7. Large-Scale CRM197 Expression and Purification

The development of a large-scale protein production system requires a transition from expression in shake flasks to a high-cell-density cultivation in a fermenter. The fed-batch fermentation protocol was adapted from [14]. A detail to emphasize is the decreased temperature: 28 °C during the initial culture growth and 23 °C after the induction of CRM197 expression. The induction phase lasted for 12 h; then, the biomass was harvested. The yield was 135 g of wet biomass per 1 L of the final cell culture.

The CRM197 purification from the soluble protein fraction was performed by ultrafiltration, three consecutive chromatographic steps, and a final ultrafiltration. The yield of purified monomeric CRM197 was in the range of 1.1–2.0 mg per 1 g of wet biomass, which corresponded to 150–270 mg per 1 liter of cell culture.

Importantly, to calculate this yield, the amount of CRM197 obtained after the three chromatographic steps was normalized to the biomass amount taken for the purification batch. The amount of purified CRM197 was inevitably lower than the amount of “soluble CRM197” in the initial cell lysate. However, the resulting yield obtained using the high-cell-density cultivation in the fermenter was an order of magnitude higher than the best results obtained during cultivation in test tubes or shake flasks.

### 3.8. Analysis of the Purified CRM197

The purified CRM197 was examined using Laemmli electrophoresis under reducing conditions (Figure 4A) and Western blotting (Figure 4B). The proteolytic breakdown level of the CRM197 molecules was below 6.25%, according to the gel densitometry data, although some low quantities of the free A subunit could be detected by Western blotting with MBS5305972 antibodies (specific to the A subunit). The purified CRM197 was homogeneous, according to the native PAGE data (Figure 4C). At the same time, semi-purified CRM197 showed a significant level of heterogeneity on the native gel. A significant proportion of CRM197 was unable to bind to the Capto Q resin. The corresponding protein sample mainly consisted of the CRM197 band on the SDS gel, which formed a smear on the native gel at the same time. This fact indicated the presence of misfolded CRM197 molecules and aggregates as well as complexes with host cell proteins and nucleic acids. At the same time, the Capto Q eluate on the native gel demonstrated several focused protein bands, indicating the removal of the CRM197 misfolded forms. The second and third chromatographic steps allowed the removal of the CRM197 forms with altered mobility on the native gel.

The fully purified CRM197 was found to be monomeric, according to analytical size-exclusion chromatography (Figure 4D,E), with an apparent molecular mass of 55.7 kDa, which was in good agreement with the theoretical value of 58.5 kDa. The final CRM197 solution contained 2.6% oligomers and 4.6% shorter fragments. Importantly, no high-molecular-weight aggregates were detected in the void volume. Later, the purified CRM197 solution was concentrated up to 22 mg/mL without visible aggregation or a reduction in solubility.

According to the N-terminal protein sequencing, 70% of the CRM197 molecules carried an additional N-terminal Met residue, while the next nine amino acid residues corresponded to the original CRM197 sequence. The rest of the molecules lacked the N-terminal Met residue, presumably due to the activity of *E. coli* methionine aminopeptidase [44]. A similar heterogeneity in the CRM197 primary structure was reported earlier [19]. It did not lead to any further heterogeneity. The secondary and tertiary structures of the CRM197 preparation with the mixture of the N-terminal variants were completely the same as those of the original CRM197 expressed in *C. diphtheriae* without the additional N-terminal Met residue [27].

## 4. Discussion

Due to a large worldwide demand for CRM197, many research teams have attempted to produce the recombinant monomeric CRM197 in reasonable quantities utilizing very divergent expression protocols and various hosts. Those reported in peer-reviewed articles are listed in Table 2 and compared to the expression system described in the present work.

Despite the high levels of CRM197 biosynthesis that are claimed, the presently available protocols yield either insoluble CRM197 [11,16,17] or tagged CRM197 variants [11,12,16,18] that are not suitable for vaccine synthesis without additional processing. We have shown that CRM197 may be easily refolded to the soluble state, but the resulting protein form is not monomeric; it fails to bind to the ion-exchange chromatography resin. Tag removal is a very resource-consuming process, making the tagged variant of CRM197 very cost-inefficient. Residual tagged CRM197, inevitably present in the processed CRM197 after the tag removal, also adds the risk of tag-specific antibody generation if such CRM197 is used in vaccines.

Although most works have relied on *E. coli*, some protocols based upon other host organisms were also developed. The CRM197 expression in Gram-positive *Bacillus subtilis* clearly suffered from low yield [21]. The expression in the methylotrophic yeast *Pichia pastoris* provided a final yield of 113 mg/L, which was comparable to the productivity of bioprocesses based on *C. diphtheriae* [25]. Importantly, CRM197 from *P. pastoris* was partially *N*-glycosylated, as confirmed by peptide-*N*-glycosidase F (PNGase F) treatment [25]. The glycosylation of CRM197 is highly undesirable because it can decrease conjugation efficiency, shield some epitopes, and induce an immune response toward the *N*-glycans [45,46,47]. The CRM197 expression in Gram-negative *P. fluorescens* was successful enough to use the created system at an industrial scale, although the exact yield was not reported.

Among the protocols based on *E. coli*, the most impressive CRM197 yield was achieved using its translocation into the periplasm [14]: the value of 3 g/L was an order of magnitude greater than all other results (Table 2). However, the article [14] lacked any description of the CRM197 purification procedure. There were also no size-exclusion chromatography data for the purified CRM197. Moreover, a large discrepancy was observed between the CRM197 quantifications using ELISA and gel densitometry. The CRM197 concentration in the periplasmic extract determined using ELISA was much higher than the CRM197 concentration determined using SDS-PAGE densitometry. This difference may be the result of CRM197 non-covalent multimerization greatly enhancing its reaction with antibodies. In SDS-PAGE, these noncovalent multimers are dissociated by the SDS sample treatment. According to our data obtained for the periplasm-targeted CRM197, the target protein mainly failed to bind to the chromatography resin, and the yield of purified CRM197 was relatively low.

It should be emphasized that the “CRM197 yield” is given in various works using different quantification techniques and different samples—cell lysates, protein fractions, or purified protein (see Table 2 footnotes). The yield reported in the present article was calculated as the amount of soluble CRM197 obtained at the end of the purification process, normalized to the weight of the biomass used for the purification. According to our measurements, even for the cytoplasmic expression at a low temperature, about a half of the soluble CRM197 in the cell lysate did not bind to the anion-exchange resin, indicating the presence of misfolded or multimerized CRM197 molecules. Therefore, assessing the amount of “soluble CRM197” is acceptable when optimizing expression conditions, but only “purified monomeric CRM197” should be reported as the actual final yield of any expression and purification protocol.

Besides the obtained yield, one more important feature of a protein production protocol is the fermentation scale. The transition from shake flasks to a fermenter is usually poorly predictable, while the fed-batch fermentation is linearly scalable [14], with only minor process changes from the benchtop vessels to the industrial-scale fermentation setups. Thus, only those protocols that have been performed in fermenters may be considered mature enough for industrial applications (see Table 2).

CRM197 expression in the cytoplasm of specifically enhanced *E. coli* strains has been attempted several times with various results. The Origami strain contains knockouts of the *trxB* and *gor* genes coding for thioredoxin reductase and glutathione reductase [48,49] and a compensating mutation in the gene encoding AhpC* peroxidase, which acquires the activity of disulfide bond reductase [50]. Using the Origami B (DE3) strain, along with the overexpression of additional chaperones (GroES, GroEL, DnaK, DnaJ, GrpE, and Tf in different combinations), allowed researchers to obtain up to 150 mg/L soluble CRM197 [12]. Importantly, Origami cells lack the capacity to correct any mis-oxidized pairs of cysteines in the cytoplasm. During further metabolic engineering, this activity was added through the overexpression of a cytoplasmic version of disulfide bond isomerase DsbC [33,40]. The resulting *E. coli* SHuffle strain was used in the present work and allowed us to obtain a higher yield of the soluble CRM197. Thus, we assume that the overexpression of *E. coli* chaperones is dispensable in the case of CRM197. The expression of the soluble CRM197 in the SHuffle strain using a specifically altered CRM197 gene was reported previously, but no data concerning the protein yield were disclosed [19].

*E. coli* is a mesophilic bacterium and grows well in the temperature range of 21–49 °C. Its optimum is around 37 °C, while the minimum for measurable growth is at 7.5 °C [51]. The temperature of 37 °C is typically used for *E. coli* cell culture growth as well as for the expression of heterologous proteins. However, when the solubility of the target protein becomes an issue, a long-known first-choice solution is lowering the expression temperature [52,53,54]. This effect is based on several different mechanisms [53]. First, a decreased temperature results in a lower metabolism and a slower protein biosynthesis, thus providing more time for the proper protein folding. Second, protein aggregation is, in general, favored at higher temperatures since hydrophobic interactions are characterized by strong temperature dependence [55]. Third, the native *E. coli* chaperones GroEL and GroES, required for the proper folding of many different proteins, are mostly active in the range of 20–30 °C [51].

This strategy was used in the present work: lowering the expression temperature increased the yield of soluble CRM197. This result was completely in agreement with the data reported by others [12,14,15,18]. Importantly, no successful protocol for CRM197 expression relied exclusively on the temperature decrease; it was rather a valuable addition to other biotechnological solutions. A question remains as to which expression temperature should be chosen, namely 30 °C [15,18], 23 °C [14], or even 15 °C [12]. Considering various expression vectors and strains, along with different growth medium compositions, one can expect that there is no universal value, and the optimal expression temperature should be determined individually in each case.

Although synthetic genes with the optimal codon usage pattern and a streamlined mRNA secondary structure may be constructed by automated online tools without any significant effort of the end-user, even small variations in the gene structure may significantly affect the gene expression level. All articles regarding CRM197 expression in *E. coli* (including the present work) report codon optimization for this organism. However, there are no common standards for such optimization and the optimized nucleotide sequences (typically not shown) could have varied in different works. Remarkably, the codon optimization itself increases the production of the target protein, but CRM197 remains insoluble. Therefore, increasing its solubility requires some additional steps.

The optimization of mRNA structure is usually performed along with codon optimization and is often not reported in detail. The present work shows that a short hairpin in the beginning of the CRM197-encoding gene can affect its expression level several-fold. The difference in one nucleotide is so obscure that its importance was not predicted during our initial sequence analysis. Only after the difference in the expression levels was detected experimentally was an explanation regarding the hairpin found. No similar effects have been discussed explicitly in published peer-reviewed articles (Table 1). However, searching for such an mRNA structure, which would not prevent effective translation, is implied in any modification of the CRM197 N-terminus, including the N-terminal His_6_ tag addition [11,16].

## 5. Conclusions

Although the heterologous expression of recombinant proteins is a routine technique in biotechnology and codon optimization tools have been widely used for decades, unpredicted outcomes are still possible. In our case, a synonymous substitution of only a single nucleotide at the beginning of the target gene resulted in a four-fold change in its expression level. This change did not correlate with the differences in free energies calculated for the supposed mRNA hairpins but inversely correlated with the probability of the extended hairpin formation. The explanation was found post factum and referred to long-distance interactions inside the full-length mRNA.

Choosing the proper codon variant allowed us to construct an *E. coli* strain capable of the effective expression of CRM197 in a soluble form. The strain was successfully cultivated under high-cell-density conditions in a fermenter. The advantages of our protocol include the high yield of the soluble CRM197 and the lack of need for CRM197 refolding or tag removal.

## Figures and Tables

**Figure 1 biotech-12-00009-f001:**
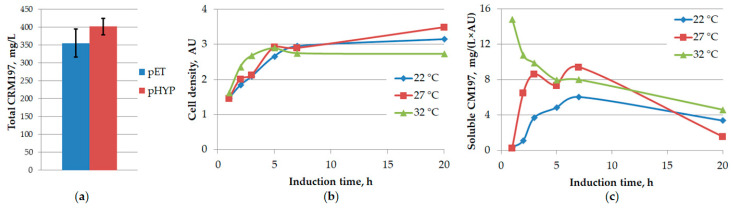
Optimization of CRM197 expression conditions. (**a**) Choosing an expression vector. Total concentration of CRM197 (soluble + insoluble) obtained upon its expression in the *E. coli* SHuffle T7 strain at 37 °C using pET28a-CRM197-gcc or pHYP-CRM197-gcc expression vectors. (**b**) *E. coli* SHuffle T7 strain growth in shake flasks at different temperatures during CRM197 expression from the pHYP-CRM197-gcc plasmid. (**c**) Concentration of soluble CRM197 changing over time at different temperatures upon CRM197 expression from the pHYP-CRM197-gcc plasmid in the *E. coli* SHuffle T7 strain in shake flasks. N = 2.

**Figure 2 biotech-12-00009-f002:**
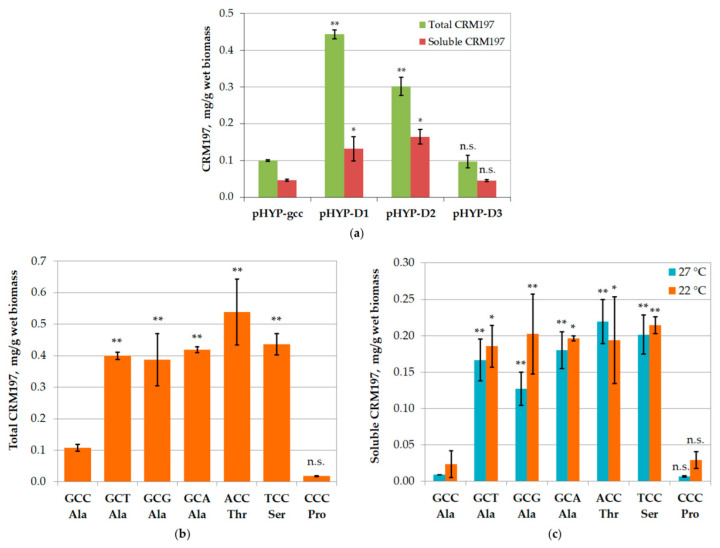
Effects of mutations in the beginning of the CRM197-encoding gene on the CRM197 expression level in the *E. coli* SHuffle T7 strain. (**a**) Short deletions. CRM197 levels after its overnight expression at 22 °C from the plasmids pHYP-D1, pHYP-D2, and pHYP-D3 in comparison with those from the pHYP-CRM197-gcc plasmid. (**b**) Point mutations. Levels of total (soluble + insoluble) CRM197 after its overnight expression at 22 °C from the plasmids pHYP-CRM197-gct, pHYP-CRM197-gca, pHYP-CRM197-gcg, pHYP-CRM197-acc, pHYP-CRM197-tcc, and pHYP-CRM197-ccc in comparison with that from the pHYP-CRM197-gcc plasmid. (**c**) Point mutations. Levels of soluble CRM197 after its overnight expression at 27 or 22 °C from the same plasmids as in panel (**b**). N = 2, *—*p* < 0.05; **—*p* < 0.01; n.s.—*p* > 0.05 (ANOVA, pairwise comparison with GCC).

**Figure 3 biotech-12-00009-f003:**
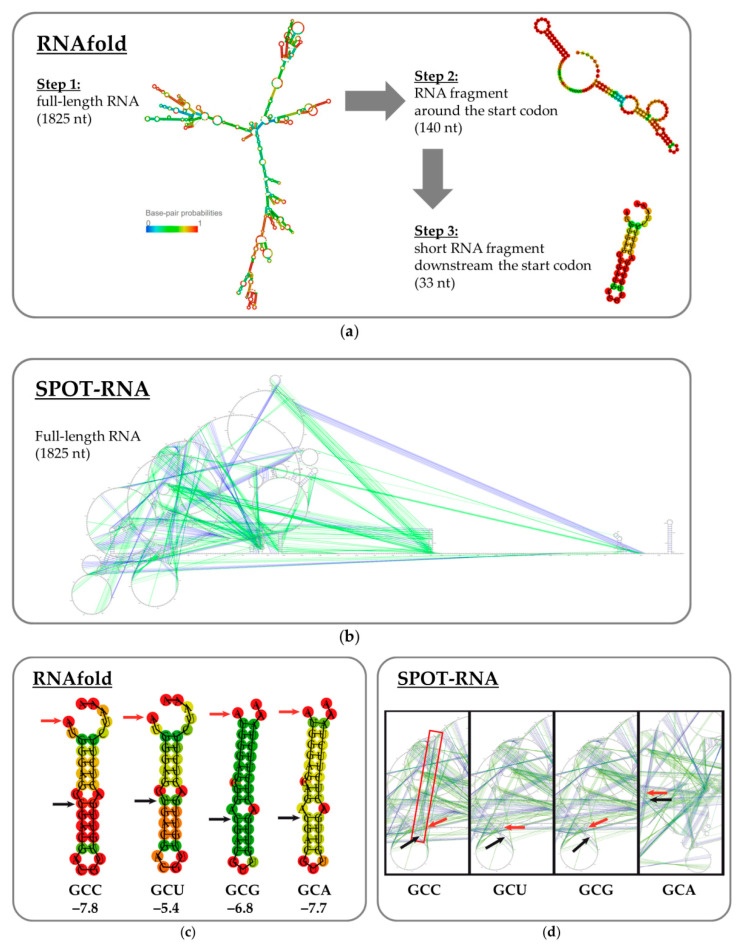
Effects of the point mutations on mRNA secondary structure predicted using the RNAfold and SPOT-RNA programs. (**a**) Selecting the mRNA fragment for the RNAfold analysis. The predicted probability of being paired or unpaired is color-coded from blue (zero probability) to red (100% probability). To see the RNAfold results for all full-length mRNA variants, please refer to Appendix A in the Appendix A. (**b**) The full-length mRNA structure predicted by SPOT-RNA (in a radial representation). The secondary structure is depicted with colored lines (blue for canonical base pairs and green for non-canonical lone pairs and triplets). (**c**) The optimal secondary structure of the hairpin in the beginning of the CRM197-encoding mRNA (33 bases) predicted by the RNAfold algorithm. Each red arrow indicates the first nucleotide of the start codon; each black arrow points to the mutant nucleotide. The variable nucleotide triplets are written near the corresponding hairpins. The minimum free energy values for the shown structures are given below in kcal/mol. The probability of hairpin formation is marked by color and increases from blue to red. (**d**) Long-range interactions in the CRM197-encoding mRNA predicted by the SPOT-RNA program. The GCC codon participates in such interactions (highlighted with a red rectangle), while the GCU and GCG codons do not. Each red arrow indicates the first nucleotide of the start codon; each black arrow points to the mutant nucleotide. Only a portion of the mRNA is shown (in a radial representation). The secondary structure is depicted with colored lines (blue for canonical base pairs and green for non-canonical lone pairs and triplets).

**Figure 4 biotech-12-00009-f004:**
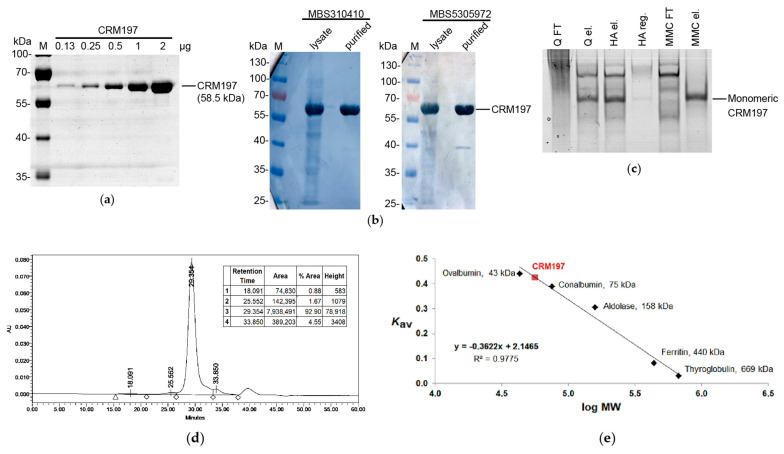
Analysis of the purified soluble CRM197. (**a**) Increasing concentrations of the purified CRM197 analyzed by Laemmli gel electrophoresis under reducing conditions. (**b**) Detection of CRM197 using Western blotting with monoclonal anti-DT antibodies MBS310410 and MBS5305972 (MyBioSource). Lanes 1 and 3, cell lysate; lanes 2 and 4, the purified CRM197. (**c**) Analysis of samples taken at different CRM197 purification stages using native gel electrophoresis. The samples are in the following order: Capto Q flowthrough, Capto Q eluate, ceramic hydroxyapatite eluate, ceramic hydroxyapatite regeneration (500 mM potassium phosphate), Capto MMC flowthrough, and Capto MMC eluate. The CRM197 band is marked with an arrow. (**d**) Size-exclusion chromatography of the purified CRM197 on the Superdex 200 HR10/300 column. (**e**) Calibration curve for the size-exclusion chromatography.

**Table 1 biotech-12-00009-t001:** CRM197 amounts in several conjugate vaccines.

Vaccine	Marketing Authorization Holder	Bacterial Polysaccharides	CRM197 Amount in One Dose, µg
Menveo	GSK Vaccines S.r.l.	*Neisseria meningitidis* polysaccharides, serogroups A, C, Y, and W-135	32.7–64.1
Menjugate	GSK Vaccines S.r.l.	*Neisseria meningitidis* polysaccharides, serogroup C (strain C11)	12.5–25.0
Prevenar 13	Pfizer Europe MA EEIG	*Streptococcus pneumoniae* polysaccharides, serotypes 1, 3, 4, 5, 6A, 6B, 7F, 9V, 14, 18C, 19A, 19F, and 23F	32
Vaxneuvance	Merck Sharp & Dohme B.V.	*Streptococcus pneumoniae* polysaccharides, serotypes 1, 3, 4, 5, 6A, 6B, 7F, 9V, 14, 18C, 19A, 19F, 22F, 23F, and 33F	30

**Table 2 biotech-12-00009-t002:** Peer-reviewed articles focused on CRM197 expression and purification.

Year	Expression Host	Compartmentalization	Solubility	Refolding Required	Protein Processing Required	Fermentation Scale	Cell Density	Protein Yield, mg/L	Reference
1983	*Corynebacterium diphtheriae*	Secreted	Soluble	No	No	5-L fermenter	Low	175–250 ^3^	[10]
1999	*Bacillus subtilis*	Secreted	Soluble	No	No	Shake flask	Low	7.1 ^3^	[21]
2011	*Escherichia coli*	Cytoplasmic	Insoluble (inclusion bodies)	Yes	Yes ^1^	Shake flask	Low	250 ± 50 ^4^	[11]
2016	*E. coli*	Cytoplasmic	Soluble + insoluble	No	Yes ^1^	Shake flask	Low	154 ± 13 ^3^	[12]
2017	*E. coli*	Periplasmic	Soluble (?) ^5^	No	No	150-L fermenter	High	3200 ^4,5^	[14]
2017	*E. coli*	Cytoplasmic	Soluble + insoluble	No	No	2-L fermenter	High	106 ± 1.5 ^3^	[15]
2018	*E. coli*	Cytoplasmic	Insoluble (inclusion bodies)	Yes	Yes ^1^	Shake flask	Low	196 ^4^	[16]
2018	*E. coli*	Cytoplasmic	Insoluble (inclusion bodies)	Yes	No	20-L fermenter	High	Not specified	[17]
2021	*Pichia pastoris*	Secreted	Soluble	No	No	16-L bioreactor	High	113 ^6^	[25]
2022	*E. coli*	Cytoplasmic	Soluble + insoluble	No	Yes ^2^	Shake flask	Low	130 ^4^	[18]
2022	*E. coli*	Cytoplasmic	Soluble	No	No	16-L fermenter	High	150–270 ^6^	This work

^1^ Tag cleavage by enterokinase is required. ^2^ SUMO tag removal is required. ^3^ The yield of soluble CRM197. ^4^ The yield of total (soluble + insoluble) CRM197. ^5^ No proof of monomeric state. ^6^ The yield of purified soluble CRM197.

**Table 3 biotech-12-00009-t003:** Plasmids and variants of the CRM197-encoding gene used in the present work ^1^.

Plasmid Name	Original Vector	Mutation Type	The Initial Codons of the Target Gene	The Initial Amino Acid Residues of the Target Protein
pET28a-CRM197-gcc	pET28a	None	ATGGGAGCCGACGAC	MGADD
pHYP-CRM197-gcc	pHYP	None	ATGGGAGCCGACGAC	MGADD
pHYP-CRM197-peri	pHYP	Leader peptide insertion	ATGATTAAATTTCTCTCTGCATTAATTCTTCTACTGGTCACGACGGCGGCTCAGGCTGGAGCCGACGAC	MIKFLSALILLLVTTAAQAGADD
pHYP-D1	pHYP	Deletion	ATG---GCCGACGAC	M-ADD
pHYP-D2	pHYP	Deletion	ATG------GACGAC	M--DD
pHYP-D3	pHYP	Deletion	ATG---------GAC	M---D
pHYP-CRM197-gca	pHYP	Synonymous substitution	ATGGGAGC**A**GACGAC	MGADD
pHYP-CRM197-gcg	pHYP	Synonymous substitution	ATGGGAGC**G**GACGAC	MGADD
pHYP-CRM197-gct	pHYP	Synonymous substitution	ATGGGAGC**T**GACGAC	MGADD
pHYP-CRM197-acc	pHYP	Nonsynonymous substitution	ATGGGA**A**CCGACGAC	MG**T**DD
pHYP-CRM197-tcc	pHYP	Nonsynonymous substitution	ATGGGA**T**CCGACGAC	MG**S**DD
pHYP-CRM197-ccc	pHYP	Nonsynonymous substitution	ATGGGA**C**CCGACGAC	MG**P**DD

^1^ Affected nucleotides and amino acid residues are underlined.

## Data Availability

All data generated or analyzed during this study are included in this article and its Appendix A.

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
