# Peer review of "High-Level Production of Soluble Cross-Reacting Material 197 in Escherichia coli Cytoplasm Due to Fine Tuning of the Target Gene’s mRNA Structure"

_biotech, 2023, doi:10.3390/biotech12010009_

Round 1

Reviewer 1 Report

The premise of the paper to evaluate the impact of RNA structure on protein expression is interesting. But there are a few major comments about this publication. 

A) The objective of any expression study is to improve the titer beyond the current values in the literature. What authors describe in this publication is just comparable to or lower than published values.

B) The authors describe in figure 2A total and soluble protein levels, but in figure 2B similar analysis is missing. As the objective is always to improve soluble protein expression, this critical data was not evaluated.

C) The authors tried lowering the temperature with the original construct. As the objective of the publication was to increase titers, authors should perform similar experiments for different variants and characterize solubility, N-terminal sequence, etc.

Minor comment

A) In figure 4, please add markers for CRM197 molecular weight. Also, report % Area for all peaks. 

Author Response

Comment A. The objective of any expression study is to improve the titer beyond the current values in the literature. What authors describe in this publication is just comparable to or lower than published values.

Reply. Numerical values of our yield are indeed comparable to several other results. However, there are some important difference in details.

First, different values are given as “yield” in different articles (see Table 2 footnotes). We report the yield of purified monomeric CRM197. The other works typically report “total CRM197” or “soluble CRM197”. As we have seen in our experiments, about one-half of “soluble CRM197” detected by SDS-PAGE (confirmed by western blot) does not bind to Capto Q column under the standard conditions. Most likely, these species are misfolded CRM197 molecules. Additionally, some amount of CRM197 is inevitably lost during further purification and buffer exchange steps. Thus, we are confident that only “purified monomeric CRM197” should be taken as a real yield.

Second, our CRM197 does not require refolding – a time-consuming procedure with a relatively low yield. We were not able to get truly soluble CRM197 by refolding, and strongly suggest that CRM197 expression in soluble form is a serious advantage.

Third, the primary structure of our CRM197 is completely identical to its natural form, except the Met[-1] residue. It contains no expression tags or additional domains. This is important in light of further certification for pharmaceutical use.

To emphasize these details, we slightly modified the data representation in Table 2 and added several phrases into Discussion.

Comment B. The authors describe in figure 2A total and soluble protein levels, but in figure 2B similar analysis is missing. As the objective is always to improve soluble protein expression, this critical data was not evaluated.

Reply. We performed some additional experiments and added the data concerning the soluble protein expression (see Fig. 2C).

Comment C. The authors tried lowering the temperature with the original construct. As the objective of the publication was to increase titers, authors should perform similar experiments for different variants and characterize solubility, N-terminal sequence, etc.

Reply. We added data concerning the soluble protein levels for all mutants at different temperatures (see Fig. 2C).

The N-terminal sequencing was not performed for all protein mutants. This time-consuming technique requires highly-purified protein, which is especially difficult to obtain for the poorly expressed mutants. The aim of the work was to assess influence of the mRNA codons onto the translation efficiency. All the codon identities were proved by DNA sequencing, while the N-terminal sequencing was performed only for the final highly purified protein sample.

Comment D. In figure 4, please add markers for CRM197 molecular weight. Also, report % Area for all peaks. 

Reply. We changed Fig. 4D accordingly.

Reviewer 2 Report

CRM197 is is a non-toxic mutant of the diphtheria toxin and is widely used as a carrier protein in conjugate vaccines, but also as a boosting antigen for immunization against diphtheria. Several strategies have been developed during the last years to overcome the low yield expression, but there is still space for further improvements.

The original expression system for CRM197 implies cultivation of corresponding Corynebacterium diphtheriae strains, which is the natural host of DT and the causative agent of diphtheria, but during the last years, alternative expression hosts have been used.

Three expression systems were successful enough to serve as a basis for a large-scale manufacturing. The resulting CRM197 proteins are distributed as PeliCRM™ (Ligand Pharmaceuticals, previously sold by Pfenex, expression in P. fluorescens) [23], EcoCRM™ (Fina BioSolutions LLC, expression in E. coli), and simply CRM197 (Scarab Genomics LLC, expression in Clean Genome® E. coli).

The aim of the work is to describe a novel CRM197-producing Escherichia coli strain, optimized by several expedients, as a fine tuning of the mRNA structure (the disruption of the single hairpin in the start codon area), to increase the CRM197 expression level several times, resulting in the 150270 mg/L (1.12.0 mg/g wet biomass) yield of the pure CRM197 protein.

Even if the aim of the work is promising, it fails to provide enough information about the novelty of this expression strategy. It is in fact already commercialized CRM197 expressed in E coli,  and author do not highlight efficiently the improvement they brought to the process. Indeed, authors themselves affirm that the presented protocol is just in line with the other strategies already used in industry.

Moreover, sometimes,  it looks like authors  have reviewed the other strategies instead of focusing on their work.

Specific comments :

Line 267: “Refolding of CRM197 from inclusion bodies” paragraph

I would just summarize in a sentence that it was not possible to recover monomeric CRM197 from inclusion bodies, as description of failed experiment would fit better with a thesis than with a scientific paper.

Line 325:

Do authors keep the deletion of two aminoacids at the N-term in the final CRM197 protocol? It is definitely not clear from the text, because the deletion of these aa leads to an improvement in soluble fraction, as this is the fraction authors will use for the purification.

On the other hand, authors explore the effect of codon changing and then assess that they will keep the GCT as “the DT gene sequence deposited in GenBank (X00703.1) contained this variant”. But in this case they assess only the total protein (figure 2B) but do not mention the effect of the mutation on the soluble fraction. I would recommend to revise this paragraph, and in case the information is missing, to evaluate the soluble fraction contribution to the total protein.

I also recommend to schematize the use ofonline tools RNAfold and SPOT-RNA, because this part of the text is really difficult to follow. If the aim of the experiment is still the optimization of CRM197 sequence for the purification, it must be easier to understand the aim of this specific experiment, and the results obtained which contribute to the final aim of the work.

Line 602:

The conclusion paragraph do not summarize and highlights the real conclusions of the work, so I would recommend to revise it and put more emphasis on the optimization of the expression protocol.

Author Response

Comment. Sometimes, it looks like authors have reviewed the other strategies instead of focusing on their work.

Reply. We shortened overview of literature data in the Discussion and added some sentences to emphasize the advantages of the present work.

Comment. Line 267: “Refolding of CRM197 from inclusion bodies” paragraph

I would just summarize in a sentence that it was not possible to recover monomeric CRM197 from inclusion bodies, as description of failed experiment would fit better with a thesis than with a scientific paper.

Reply. Refolding was described in numerous works as an appropriate method to obtain soluble CRM197 suitable for further use. We failed to reproduce this technique and suggest that CRM197 refolding is in fact non-functional. Thus, we report some experimental details which support our point of view. The statement about superiority of CRM197 expression in a soluble form would be unfounded otherwise.

Comment. Line 325: Do authors keep the deletion of two aminoacids at the N-term in the final CRM197 protocol? It is definitely not clear from the text, because the deletion of these aa leads to an improvement in soluble fraction, as this is the fraction authors will use for the purification.

Reply. There is no N-terminal deletion in any CRM197 variant with a point mutation (including the final variant with GCT codon). These data (the initial codons of the gene and the corresponding N-terminal amino acids of the protein) are summarized in Table 3.

We added a clarification into the text: “For a more detailed investigation, we constructed another set of plasmids without any deletions, but with all possible substitutions in the first and the third positions of the second triplet (Table 3).”

Comment. On the other hand, authors explore the effect of codon changing and then assess that they will keep the GCT as “the DT gene sequence deposited in GenBank (X00703.1) contained this variant”. But in this case they assess only the total protein (figure 2B) but do not mention the effect of the mutation on the soluble fraction. I would recommend to revise this paragraph, and in case the information is missing, to evaluate the soluble fraction contribution to the total protein.

Reply. We added data concerning the soluble protein levels for all mutants at different temperatures (Fig. 2C).

Comment. I also recommend to schematize the use of online tools RNAfold and SPOT-RNA, because this part of the text is really difficult to follow. If the aim of the experiment is still the optimization of CRM197 sequence for the purification, it must be easier to understand the aim of this specific experiment, and the results obtained which contribute to the final aim of the work.

Reply. We schematized the use of online tools RNAfold and SPOT-RNA in the updated Fig. 3.

Comment. Line 602: The conclusion paragraph do not summarize and highlights the real conclusions of the work, so I would recommend to revise it and put more emphasis on the optimization of the expression protocol.

Reply. We re-wrote the Conclusion section accordingly.